# The Influence of Acetate and Sodium Chloride Concentration on the Toxic Response of Electroactive Microorganisms

**DOI:** 10.3390/microorganisms13092077

**Published:** 2025-09-06

**Authors:** Fei Xing, Haiya Zhang, Shuhu Xiao, Hongbin Lu

**Affiliations:** 1State Key Laboratory of Environmental Criteria and Risk Assessment, Chinese Research Academy of Environmental Sciences, Beijing 100012, China; feixgy@126.com (F.X.); flying850612@126.com (H.Z.); 2State Environmental Protection Key Laboratory of Estuarine and Coastal Environment, Chinese Research Academy of Environmental Sciences, Beijing 100012, China; 3State Key Laboratory of Coal Conversion, Institute of Coal Chemistry, Chinese Academy of Sciences, Taiyuan 030001, China; luhbhg@163.com

**Keywords:** microbial fuel cell (MFC), toxicity, acetate, sodium chloride

## Abstract

This study discussed the influence of acetate and sodium chloride concentration on monitoring 2,4-dichlorophenol(2,4-DCP) by electroactive bacteria. The performance of the reactor was represented by power density, and the electrochemical activity was represented by redox capacity. At the same time, micro-electrodes were used to detect the redox potential between biofilms, and the changes in extracellular polymers and microbial community structure under different conditions were also explored. With acetate concentration of 1 g/L and sodium chloride concentration of 0.0125 g/L, the electroactive microorganisms were more sensitive to toxic substances and responded fast. The biofilm also evenly covered on the surface of the carrier, which aided in the diffusion of substances. Although the maximum power density monotonically increased with acetate concentration, high concentration of substrate may mask the inhibitory effect and affect the judgment of inhibitory results. The content of protein and polysaccharide increased monotonically with sodium chloride concentration. However, more polysaccharides would lead to high resistance to electron transfer. Compared to sodium chloride, the microbial content was more affected by acetate. The electroactive microorganisms had strong adaptability to salinity. In practical application, it is conducive to increase the sensitivity of MFCs to reasonably reduce the concentration of acetic acid and sodium chloride.

## 1. Introduction

Biological treatment is the core process for removing organic pollutants from industrial wastewater. However, industrial wastewater often contains a large amount of toxic and harmful pollutants, which can inhibit microorganisms [1] in the treatment system [2], cause deterioration of effluent quality, and threaten water ecological security. It has been reported that microbial fuel cells can treat wastewater containing refractory chemical compounds [3,4,5,6]. For many organic wastewaters with high concentration toxicants that are difficult to degrade, anaerobic pretreatment is necessary to reduce organic matter concentration and improve biodegradability before aerobic biological treatment, so anaerobic biological treatment units are easily affected. Therefore, the evaluation and control of anaerobic biological inhibition in wastewater deserves high attention. However, the current evaluation of anaerobic biological inhibition in wastewater at home and abroad is still limited to the direct determination of the final products of biochemical reactions, mainly including the evaluation of anaerobic methane production inhibition through monitoring methane content [7], and the evaluation of hydrolysis acidification acid production inhibition through monitoring volatile fatty acid content [7]. As is well known, when microorganisms contact with toxic substances, changes in biological metabolic activity occur before the accumulation of reaction products. Most existing evaluation methods for anaerobic biological inhibition are based on the accumulation of final products from biochemical reactions, and the evaluation results are relatively lagging. Electroactive microorganisms can directly reflect the magnitude of inhibitory effects by outputting electrical signals, and have broad research and application prospects in the field of wastewater biological inhibition evaluation. Electroactive microorganisms are currently suitable for biological toxicity testing of various wastewater [8], and can achieve toxicity evaluation of various substances such as organic matter [9], heavy metals [10,11], nitrates [12], and antibiotics [13].

When using electroactive microorganisms for inhibitory evaluation, high sensitivity is an important issue that needs to be addressed [14,15,16], and the water quality conditions of wastewater also indirectly affect the results of inhibitory evaluation. For example, the concentration of acetate [17] can affect the composition of cell extracellular polymers and even the community composition of microorganisms [18], and the concentration of sodium chloride also varied with different production processes.

Early studies have shown that the response of current generated by MFCs to substrate concentration followed a first-order equation, where a finite substrate concentration corresponded to a low current density [19]. The consumption rate of substrates and the metabolism of microorganisms were two important factors affecting microbial electricity production [20]. In most studies, the concentration of acetate as the substrate for MFCs was 12.2 mM [21]. When the concentration of acetate was greater than 100 mM, the current of MFCs no longer increased, and the substrate reached saturation concentration, which conformed to the growth kinetics of microbial saturated substrates [22,23]. A marine microbial fuel cell cultured with mixed bacteria, according to the Mono equation, had a current density close to a constant value when the concentration of acetate was greater than 0.50 mM [24]. There have been many reports on the effect of acetate concentration about electricity production, but there were few reports on its response to biological inhibition. Therefore, it is necessary to conduct research on the influence of acetate concentration on the inhibitory response to certain contaminants.

The concentration of sodium chloride affected the activity of anodic biofilm microorganisms and the electrochemical performance of MFCs biosensors [23]. The concentration of sodium chloride affected extracellular electron transfer and subsequently affects the power density of MFCs. Studies have shown that the power density and output voltage reach their maximum at a sodium chloride concentration of (1%, *w*/*v*) [25]. Although the conductivity of the solution improved with the increase in sodium chloride concentration, it did not significantly improve the electrical performance [26,27]. When the concentration of sodium chloride was 0.1 M, the power density of *Geobacter* spp. will exhibit significant disturbance [28]. Adding 0.1 M or more sodium chloride can alter the species presence of bacteria in the anode biofilm and ultimately reduce the power generation performance of MFCs [29]. The effect of sodium chloride concentration on the electricity generation of MFCs has been reported, but there are few reports on the biological inhibitory response, as well as the impact on biological activity and redox capacity. Therefore, it is necessary to conduct research on the biological inhibitory response of different sodium chloride concentrations to control the influent sodium chloride concentration and ensure the reliability of the inhibitory response results.

Therefore, it is necessary to conduct research on the influence of acetate and sodium chloride concentration on the inhibitory response of a microbial fuel cell during the treatment of contaminated aqueous effluents. By evaluating the performance, electrode activity, redox potential between biofilms, and extracellular polymers of MFCs, combined with microbial community structure analysis, this paper discusses the impact of water quality on the inhibitory response of electroactive microorganisms. The influence of water quality conditions on the toxicity of electroactive microorganisms has been studied to improve the reliability of sensors.

## 2. Materials and Methods

### 2.1. Construction and Operation of Microbial Fuel Cells

A dual chamber microbial fuel cell with biological anode and chemical cathode was constructed and operated as mentioned before [30]. A collection system (Xinwei 4008, China) was used to record signals as current and voltage. In three cycles, when the voltage signal became 610 ± 15 mV (mean ± SD)with 2% accuracy, the reactors were suitable for toxicity test at room temperature (20 ± 3 °C). 2,4-dichlorophenol(2,4-DCP) was added in the anode, then we recorded the time. When the voltage changed by 5%, it was considered that the electroactive microorganisms had responded to toxic substances. When the voltage change amplitude was less than 5%, it was considered that the toxic reaction had terminated. Inhibition reaction time referred to the period from the start of adding toxic substances to a voltage change amplitude of less than 5%.Inhibition rate refers to the difference in average current change between blank and adding toxic substances within the same reaction time. Appendix A shows images of an anode under different acetate concentration of 0.5 g/L, 1 g/L, and 5 g/L and sodium chloride concentration of 0.005 g/L, 0.0125 g/L, and 5 g/L.

### 2.2. Electrochemical Tests

When the MFCs were stable, the performance of the MFCs could be monitored. The power density was calculated by polarization curve with series resistances [31]. The cyclic voltammetry (CV) [32] plots were obtained by scanning from −0.6 V to +0.6 V at a 10 mV/s rate with an electrochemical workstation (CorrTest CS350, Wuhan, China).

The oxidation-reduction potential (ORP) of anode biofilm can be in situ determined by micro-electrode [33,34]. When the voltage of MFCs under different culture conditions was stable, the anode biofilm ORP gradient can be investigated with an ORP micro-electrode (Unisense RD10-9145, Copenhagen, Denmark). The micro-electrode was connected to an 8-channel multimeter (Unisense A/S, Copenhagen, Denmark), combing with an Ag/AgCl reference electrode (Unisense REF-10, Copenhagen, Denmark). The test step size was set at 10 μm after the ORP micro-electrode closed to the biofilm.

### 2.3. Analysis of Microbial Characteristics

The small pieces (0.5 cm × 0.5 cm) of carbon felt with biofilms were rinsed in sterile phosphate buffer solution (PBS) (pH = 7.1)., then stained with LIVE/DEAD Bacterial Viability Kit (Thermo Fisher Scientific 1910798, Waltham, MA, USA) for 20 min in the dark [35]. Before observing, sterile PBS was used to wash the dye. A confocal laser scanning microscope (CLSM) (Leica sp8, Wetzlar, Germany) [36] was used to observe the morphology of biofilms (six CLSM images).

The protein of extracellular polymeric substances(EPS) was extracted and investigated by a protein assay (Thermo Scientific 23227, Waltham, MA, USA) [37]. Phenol–sulfuric acid method [38] was used to evaluate the polysaccharide level. The Tianamp Bacteria DNA kit was used to evaluate DNA (Tiangen, Shanghai, China) [37].

The anodes biofilms DNA was extracted to analyze changes of microbial communities in different culture conditions. The primers of V3-V4 regions as 515F (GTGCCAGCMGCCGCGG) were amplified of the bacteria *16S rRNA* gene [39]. The content of anode electroactive bacteria was analyzed at the level of phylum in Appendix A.

## 3. Results and Discussion

### 3.1. Toxic Response on MFCs

#### 3.1.1. Effect of Acetate Concentration

The inhibition responses of electroactive microbes to 2,4-dichlorophenol(2,4-DCP) at different acetate concentrations are shown in Figure 1a, with acetate concentrations of 0.5 g/L, 1 g/L, and 5 g/L. When the acetate concentration was 0.5 g/L, the voltage could be maintained at 605 ± 10 mV for about 7.8 h without 2,4-DCP. With 1.0 g/L acetate, the voltage could be maintained at 616 ± 6 mV for about 21 h. At 5 g/L, the voltage could last for more than 25 h with 660 ± 3 mV. With the increase in acetate concentration, the voltage increased and the matrix needed a long time to deplete.

With 50 mg/L 2,4-DCP, the order of voltage decrease was 1 g/L, 0.5 g/L, and 5 g/L acetate concentrations. When the acetate concentration was 0.5 g/L, the voltage changed by 2.8% in 5 h, 79.2% in the next 12 h compared with the initial voltage 529 mV, and finally decreased to 1.8 mV at 20.5 h. With 1 g/L acetate, the voltage changed by 5.8% within 5.5 h and stabilized at 1.8 mV at 10.1 h. When the concentration of acetate was 5 g/L, the voltage did not show an obvious downward trend within 20 h. This indicated that a high concentration of substrate may mask the inhibitory effect and affect the judgment of inhibitory results. Acetate was converted into acetyl-CoA by acetyl-CoA synthase, which served as a key metabolic intermediate in the TCA cycle, promoting ATP production and carbon skeleton supply. With abundant nutrients, the ions of nutrients surrounding microorganisms were more than those of toxic substances, so microorganisms uptake more nutrients, thus masking the toxicity response. When acetate was insufficient, the production of acetyl-CoA decreased. The TCA cycle was disrupted, and the supply of oxaloacetic acid was insufficient, inhibiting energy production and amino acid synthesis. Microorganisms mainly devoted their energy to maintaining their normal growth and metabolism; thus, their response to toxic substances slowed down. When the concentration of sodium acetate was 1 g/L, the response to toxic substances was the fastest.

#### 3.1.2. Effect of Sodium Chloride Concentration

The inhibitory response of electroactive microbes to 2,4-DCP at different concentrations of sodium chloride is shown in Figure 1b, with the sodium chloride concentration of 0.005 g/L, 0.0125 g/L, and 5 g/L. Without 2,4-DCP, when the concentration of sodium chloride was 0.005 g/L, the voltage could be maintained at 654 ± 10 mV for about 8 h. For 0.0125 g/L sodium chloride, the voltage could be maintained at 616 ± 6 mV for about 21 h. When the concentration of sodium chloride was 5 g/L, the voltage could be maintained at 660 ± 10 mV for about 6.8 h. Too low or too high a concentration of sodium chloride will shorten the duration of stable voltage. Low concentration sodium chloride cannot effectively maintain the osmotic pressure of microorganisms, while high concentration of sodium chloride can lead to excessive osmotic pressure of microorganisms. Excessive or insufficient concentration of sodium chloride can affect the growth of electroactive microorganisms, and can easily lead to inaccurate results in such cases.

After adding 50 mg/L 2,4-DCP, the inhibition response time of 0.005 g/L, 0.0125 g/L, and 5 g/L sodium chloride was 5.4 h, 4.8 h, and 4.4 h, respectively. Microorganisms were prone to swelling with low sodium chloride concentration. Under high sodium chloride concentration, the effect of osmotic pressure on microorganisms was greater than that of the inhibition effect.

The average current inhibition rate [30] of electroactive microorganisms at different acetate concentrations is shown in Figure 1c. The inhibition rates of 0.5 g/L, 1 g/L, and 5 g/L acetate were 39.6%, 45.1%, and 0.3%, respectively. When the acetate concentration was 0.5 g/L, the decrease in voltage was not only due to the inhibition of 2,4-DCP, but also due to the insufficient nutrients. When the acetate concentration increased to 1 g/L, the biological inhibition response was faster, while the biological inhibition was not obvious with 5 g/L acetate, resulting in misjudgment of the inhibition results. In conclusion, acetate concentration was very important for the judgment of biological inhibition. Therefore, when the acetate concentration was 1 g/L, the electroactive microorganisms were more sensitive to toxic substances and responded fast.

The average current inhibition rate of electroactive microorganisms at different sodium chloride concentrations is shown in Figure 1d. The inhibition rates of sodium chloride concentrations of 0.005 g/L, 0.0125 g/L, and 5 g/L were 32.7%, 45.1%, and 39.9%, respectively. When the concentration of sodium chloride was too low, the microorganisms had insufficient motivation to maintain life activities, and the inhibition response to 2,4-DCP was low. With the increased concentration of sodium chloride, the inhibition rates became high. When the concentration of sodium chloride was too high, the driving force of microorganisms to maintain osmotic pressure balance was greater than that of inhibiting response, and the inhibition response result was low. A too high or too low concentration of sodium chloride will cause low results of biological inhibition. Therefore, when the concentration of sodium chloride was 0.0125 g/L, the electroactive microorganisms were more sensitive to toxic substances and responded faster.

### 3.2. Electrochemical Performance of MFCs Under Different Water Quality Conditions

#### 3.2.1. Effect of Acetate and Sodium Chloride on Power Density of MFCs

The response of electroactive microorganisms to toxic substances varied under different water quality conditions. Studying the changes in their performance was beneficial for revealing the influence of water quality conditions on inhibitory response. The power density of MFCs under different acetate concentrations is shown in Figure 2a. The maximum power densities of MFCs with acetate concentrations of 0.5 g/L, 1 g/L, and 5 g/L were 492.2 ± 84.8 mW/m^2^, 1131.3 ± 127.6 mW/m^2^, and 1592.0 ± 68.6 mW/m^2^, respectively. Adding 1 g/L sodium acetate, the performance of MFCs was stable [40].The maximum power density monotonically increased with the increase in acetate concentration. When the acetate concentration increased from 0.5 g/L to 1 g/L, the acetate concentration changed by 2 times, and the maximum power density changed by 2.3 times. When the acetate concentration increased from 1 g/L to 5 g/L, the acetate concentration changed by 5 times, and the maximum power density changed by 1.4 times. Excessive concentration of acetate had little contribution to the improvement of power density.

The power density of MFCs under different sodium chloride concentrations is shown in Figure 2b. The maximum power densities of MFCs with sodium chloride concentrations of 0.005 g/L, 0.0125 g/L, and 5 g/L were 604.1 ± 144.6 mW/m^2^, 825.8 ± 35.6 mW/m^2^, and 459.1 ± 38.3 mW/m^2^, respectively. When the concentration of sodium chloride was higher than 5.9 g/L, the activity of microorganisms was interfered [15]. When the concentration of sodium chloride increased from 0.005 g/L to 0.0125 g/L, the concentration of sodium chloride increased by 2.5 times, and the maximum power density increased by 1.4 times. When the concentration of sodium chloride increased from 0.0125 g/L to 5 g/L, the concentration of sodium chloride increased by 400 times, and the maximum power density decreased by 44%. The concentration of sodium chloride has a great influence on the power density. The higher the concentration of sodium chloride was, the smaller the power density became. When the concentration of sodium chloride was higher than 5 g/L, the power density was interfered.

#### 3.2.2. Effect of Acetate and Sodium Chloride on Electrochemical Activity of MFCs

The redox activity of biofilm can be characterized by cyclic voltammetry [32,36]. The electrochemical activities of the electrodes at different acetate concentrations are shown in Figure 2c. The redox peaks were in the range of −0.40 V to −0.25 V. When acetate concentrations were 0.5 g/L, 1 g/L, and 5 g/L, the peak current densities were 0.4 mA/cm^2^, 5.6 mA/cm^2^, and 5.2 mA/cm^2^, respectively, and the corresponding potentials were −0.32 V, −0.28 V, and −0.37 V (vs. Ag/AgCl), respectively.

When the acetate concentration increased from 2 g/L to 5 g/L, the peak current density shifted from −0.28 V to −0.37 V. The study showed that the change of the peak was related to the change of the extracellular electron transfer medium [41]. The acetate concentration had a great influence on the electron transfer medium. Increasing the acetate concentration, in addition to the cytochrome *c* [41,42] of directly transferred electrons, the content of other electron transfer mediators that may not be dominant increased, so the power density changed greatly. When the concentration of organic matter bottomed out to 0.5 g/L, it was not conducive for electron transfer because of the lack of power. The current density of 1 g/L acetate concentration was obviously higher than that of 0.5 g/L. The current density changed little, when acetate concentration increased to 5 g/L. The electrochemical activity of the anode was not unlimitedly increased with the increase in acetate concentration. Excessive increase in acetate concentration had little effect on improving the anodic redox capacity.

The electrochemical activity of the electrode at different sodium chloride concentrations is shown in Figure 2d, and the redox peak was in the range of −0.41 V~−0.36 V. When the concentration of sodium chloride was 0.005 g/L, 0.0125 g/L, and 5 g/L, the peak current densities were 2.2 mA/cm^2^, 3.0 mA/cm^2^ and 1.4 mA/cm^2^, respectively, and the corresponding potentials were −0.37 V, −0.41 V, and −0.35 V (vs. Ag/AgCl), respectively. The concentration of sodium chloride had an effect on the electron transport medium, and the change of the peak was related to the change of the extracellular electron transport medium [41]. As the concentration of sodium chloride increased, the maximum peak showed a trend of increasing and then decreasing, indicating that high sodium chloride concentration was not conducive to the transfer of extracellular electrons. Excessive sodium chloride concentration, such as 5 g/L, lead to a decrease in peak value, which was consistent with the results of power density. The electrochemical activity of the anode decreased when the concentration of sodium chloride was too high.

#### 3.2.3. Effect of Acetate and Sodium Chloride on Redox Potential Between Biofilms

The anode oxidation-reduction potential was closely related to redox ability. A micro-electrode was used to monitor the oxidation reduction potential (ORP) [43]. It was found that the redox potential decreased from the outermost layer. The redox potentials between biofilms at different acetate concentrations are shown in Figure 2e. When the acetate concentration was 0.5 g/L, ORP decreased from −269.2 ± 0.1 mV to −331.0 ± 0.1 mV, and changed by 62 mV in the range of 200 μm. The ORP value was low, and anaerobic bacteria such as methanogens were easy to breed. At 1.0 g/L, ORP decreased from −168.9 ± 3.5 mV to −209.3 ± 3.4 mV, and changed by 40 mV in the range of 200 μm. The distribution difference of ORP values between biofilms was relatively small, which can ensure the efficient utilization of organic matter. With 5.0 g/L acetate concentration, ORP decreased from −36.1 ± 19.8 mV to −206.3 ± 0.6 mV, and changed by 170 mV in 200 μm. The ORP changed in a wide range, the oxidation ability was too high, the suitable living environment of microorganisms was destroyed, and the microbial community structure was easy to change, which was not conducive for response. Overall, the redox potential began to decrease from the outside of the biofilm, the same as reported [44]. Different concentrations of sodium acetate had a significant impact on the ORP between biofilms, easily leading to changes in microbial community structure and differences in toxicity response.

The redox potential between biofilms at different sodium chloride concentrations is shown in Figure 2f. The redox potential began to decrease from the outside of the biofilm, the same trend as sodium acetate. When the concentration of sodium chloride was 0.005 g/L, ORP decreased from −55.3 ± 1.5 mV to −149.1 ± 3.0 mV, and it changed by 94 mV in the range of 200 μm, when the oxidation ability was highest. At 0.0125 g/L, ORP decreased from −97.0 ± 1.1 mV to −238.4 ± 1.4 mV, and changed by 141 mV in 200 μm. With 5 g/L sodium chloride concentration, ORP decreased from −323.8 ± 0.2 mV to −344.4 ± 2.1 mV, and changed by 20 mV. The ORP changes between biofilms were relatively small in high salinity environments, with a basic range of −300 mV, and the redox ability of biofilms was poor.

#### 3.2.4. Effect of Acetate and Sodium Chloride on Microbial Morphology of Anodes

The microbial morphology on the surface of carbon felt with different acetate concentrations is shown in Figure 3a–c. The microbial richness and biofilm thickness were different under different acetate concentrations. When the acetate concentration was 0.5 g/L, the biofilm was unevenly distributed on the surface of the carrier and the biomass was small. When acetate concentration was 1 g/L, the biofilm was evenly covered on the surface of the carrier. At 5 g/L, the biofilm was thicker and fully covered on the surface of the carrier. The biofilm thickness on the carrier surface increased with acetate concentration. The distribution and thickness of biofilms were key factors affecting the efficiency of toxicity sensing.

The microbial morphology on the surface of carbon felts with different sodium chloride concentrations is shown in Figure 3d–f. When the concentration of sodium chloride was 0.005 g/L, the biofilm was flocculently and thickly distributed on the surface of carbon fiber. When the sodium chloride concentration was low, the osmotic pressure became low. Microbial cells would expand, resulting in a loose distribution of microorganisms. At 0.0125 g/L, the biofilm was evenly covered with a thin layer on the surface of carbon fiber. When the concentration went up to 5 g/L, the biofilm aggregated in clusters on the surface of carbon fiber. At high sodium chloride concentration, high osmotic pressure can affect the normal metabolism of microorganisms, affecting material transport and energy metabolism, and inhibiting enzyme activity.

#### 3.2.5. Effect of Acetate and Sodium Chloride on Microbial Extracellular Polymer

The relationship between water quality conditions and toxicity response was closely related. The changes in the content of extracellular polymers outside the microorganisms directly affect the substances’ transfer speed. Therefore, the changes in EPS components have been studied. The content of microbial extracellular polymers under different acetate concentrations are shown in Figure 4a. EPS contained polysaccharides, proteins, and DNA, of which protein content was the highest. The protein contents of 0.5 g/L, 1 g/L, and 5 g/L acetate were 42.5 ± 8.4 mg/g, 68.0 ± 6.1 mg/g, and 116.4 ± 7.9 mg/g, respectively. The polysaccharide contents of 0.5 g/L, 1 g/L, and 5 g/L acetate were 24.4 ± 6.2 mg/g, 26.4 ± 5.1 mg/g, and 56.0 ± 16.4 mg/g, respectively. As the acetate concentration increased, the content of polysaccharides and proteins also increased. The ratio of protein to polysaccharide content at different acetate concentrations was 1.8, 2.6, and 2.0, respectively. Although the content of protein and polysaccharide increased with the increase in acetate concentration, the protein/polysaccharide ratio decreased when the acetate concentration was too high. Excessive polysaccharides can increase the resistance to electron transfer. Therefore, a too low or too high acetate concentration was not conducive to response. The inhibitory effect would be covered up with high acetate concentration, resulting in misjudgment of inhibition results.

The change of microbial extracellular polymer content under different sodium chloride concentrations is shown in Figure 4b. Among EPS contains, protein was the highest. The protein contents of 0.005 g/L, 0.0125 g/L, and 5 g/L sodium chloride were 71.4 ± 2.0 mg/g, 75.0 ± 5.6 mg/g, and 93.8 ± 8.3 mg/g, respectively. The polysaccharide contents of 0.005 g/L, 0.0125 g/L, and 5 g/L sodium chloride were 28.8 ± 3.7 mg/g, 27.4 ± 4.7 mg/g and 81.5 ± 3.0 mg/g, respectively. The content of protein and polysaccharide increased monotonically with the increase in sodium chloride concentration. For example, the content of polysaccharide in 5 g/L was 2.8 times higher than that in 0.005 g/L. The content of protein in 5 g/L was 1.3 times higher than that in 0.005 g/L. This may be one of the reasons for the decrease in MFCs performance. The ratio of protein to polysaccharide content at different substrate concentrations was 2.5, 2.7, and 1.2, respectively. The ratio of protein to polysaccharide at 0.005 g/L was 2.1 times higher than that at 5 g/L. In addition to osmotic pressure, the effect of sodium chloride concentration on microorganisms also affected the polysaccharide content of extracellular polymers. When the concentration of sodium chloride was high, the microorganisms secreted more polysaccharides, and the response was slowed down because of high electron transfer resistance.

#### 3.2.6. Effect of Acetate and Sodium Chloride on Microbial Community Structure

In this study, the microorganisms in MFCs were mixed bacteria, and the electroactive microorganisms were the main bacteria. Studies have shown that formaldehyde and chloramphenicol [45] can affect the anode microbial community structure. Compared with the control group, the relative abundance of *Geobacter* spp. decreased from 81% to 53% when 1 mg/L formaldehyde was added [18]. When formaldehyde concentration increased from 63 mg/L to 169 mg/L, the relative abundance of *Geobacter* spp. decreased from 59% to 44% [46]. This study discussed the effects of different water quality conditions on microbial community structure.

The co-occurrence network diagram between microbial community and water quality conditions is shown in Figure 5a. The water conditions included acetate concentration and sodium chloride concentration. The concentrations of acetate were 0.5 g/L, 1 g/L, and 5 g/L, respectively. The concentrations of sodium chloride were 0.005 g/L, 0.0125 g/L, and 5 g/L. The orange dots in the figure represent species at the genus level. Different conditions in the intermediate zone were associated with microorganisms. The microorganisms affected by different conditions were *Geobacter*, *Desulfomonile*, *Azoarcus*, *Gordonia*, *Rhodobacteraceae*, *Devosia*, *Shinella*, *Rhodococcus*, *Flavobacterium*, *unclassified_f__Rhodobacteraceae*, *Chryseobacterium*, *Acidovorax*, *Petrimonas*, *Christensenellaceae_R-7_group*, *Fusibacter*, *Proteiniphilum*, *Anaerovorax*, and so on. Among them, *Geobacter* and *Azoarcus* were electrogenic bacteria reported in many studies [30].

The proportion of the dominant species composition for each sample is shown in Figure 5b. There were *Geobacter*, *Proteiniphilum***,**
*norank_f_Synergistaceae*, *Azoarcus*, *Gordonia*, *Flavobacterium*, *Fusibacter*, *Chryseobacterium*, *unclassified_o_Chitinophagales*, and *Christensenellaceae_R-7_group*. *Proteinophilumis* was an anaerobic acetophilic bacterium that produced acetic acid. *Azoarcus* was a bacterium belonging to the genus Azovibrio, which helped to remove nitrogen pollutants from wastewater and improve water quality. *Norank_f_Synergistackeae*, as a methane producing bacterium, played an important role in anaerobic digestion processes. *Gordonia* was a potential organic pollutant degrading bacterium. *Unclassified_o_chitinophagales* played a role in biodegradation and biotransformation processes. *Flavobacteria* can degrade organic matter and participate in carbon and nitrogen cycles, which play great significance for environmental remediation and pollution control. Microorganisms can transmit signals and interacted with each other, so further research can be conducted on their interactions. The content of *Geobacter*, *Flavobacterium*, *Fluviicola*, and *Unclassified_o_chitinophagales* increased first and then decreased with the concentration of sodium acetate. This was consistent with the trend of the redox ability of MFCs. The content of *Proteinophilum*, *Norank_f_Synergistackeae*, and *Christensenellaceae_R-7_group* increased with the increase in sodium acetate concentration. This was consistent with the overall trend of power density changes exhibited by MFCs. The content of *Gordonia*, *Azoarcus*, and *Chryseobacterium* decreased with sodium acetate concentration. Except for *Azoarcus*, the content of other bacteria did not change significantly with the increase in sodium chloride concentration. Compared to sodium chloride, the microbial content was more affected by acetate.

The similarity and difference between the communities of the control group and the treatment group at genus level are shown in Figure 5c. The closer the two sample points were, the more similar the composition of the two samples was. The low, medium, and high concentration of sodium acetate had a significant impact on the microbial community structure. The electroactive microorganisms had strong adaptability to salinity, and the impact of a 1000-fold change in sodium chloride concentration on microbial community structure was relatively small.

For the mixed bacteria system, electrons could be exchanged between bacteria, not only *Geobacter* spp., and the transfer of electrons was the result of the interaction between bacteria [47]. Therefore, appropriate concentration of the acetate and sodium chloride in the mixed bacteria system can increase the relative content of electroactive microorganisms. It is conducive to biological inhibitory response and can improve the sensitivity of the sensor.

## 4. Conclusions

The impact of water quality conditions, such as acetate and sodium chloride concentration, on monitoring toxic substances by electroactive bacteria, was investigated. The concentration of acetate had a significant impact on the community structure of microorganisms. High acetate concentration increased the overall power density of the sensor, but it reduced the redox ability of biofilm, leading to a decrease in the sensitivity of electroactive microorganisms, masking their toxic effects and causing misjudgment. The change in sodium chloride concentration mainly affected the physiological characteristics of microorganisms through osmotic pressure. High concentrations of sodium chloride can promote the extracellular secretion of more polysaccharides by microorganisms, increase mass transfer resistance, and reduce toxicity response efficiency. Extreme conditions have a significant impact on the distribution of microbial communities, and the interaction between microorganisms needs further investigation. It was helpful for electroactive microorganisms to maintain sensitivity with lower acetate and sodium chloride concentration. In practical application, the range of water quality warning needs to be jointly determined by luminescent bacteria and other methods, when the MFCs are applied in warning of biological treatment units of sewage treatment plants.

## Figures and Tables

**Figure 1 microorganisms-13-02077-f001:**
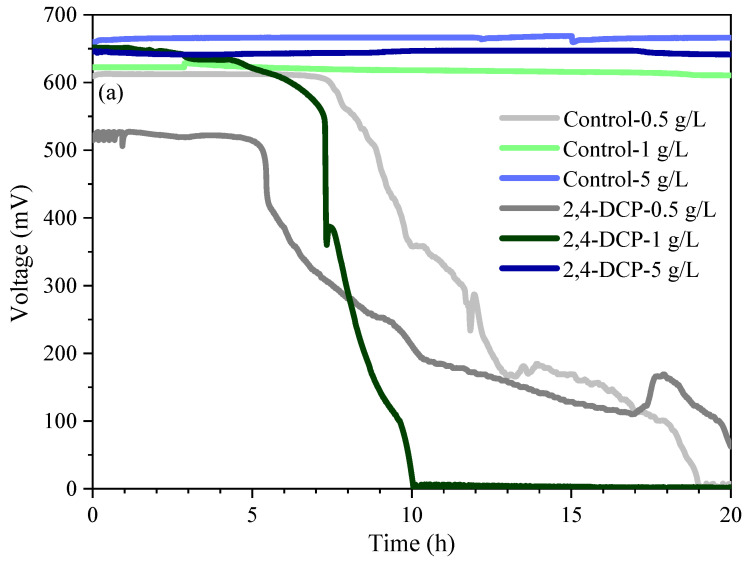
Profile of voltage with 2,4-DCP under different acetate concentration and sodium chloride concentration. (**a**) acetate concentration of 0.5 g/L, 1 g/L, and 5 g/L; (**b**) sodium chloride concentration of 0.005 g/L, 0.0125 g/L, and 5 g/L; (**c**) the inhibition rates of 2,4-DCP under different acetate concentration; (**d**) the inhibition rates of 2,4-DCP under different sodium chloride concentration.

**Figure 2 microorganisms-13-02077-f002:**
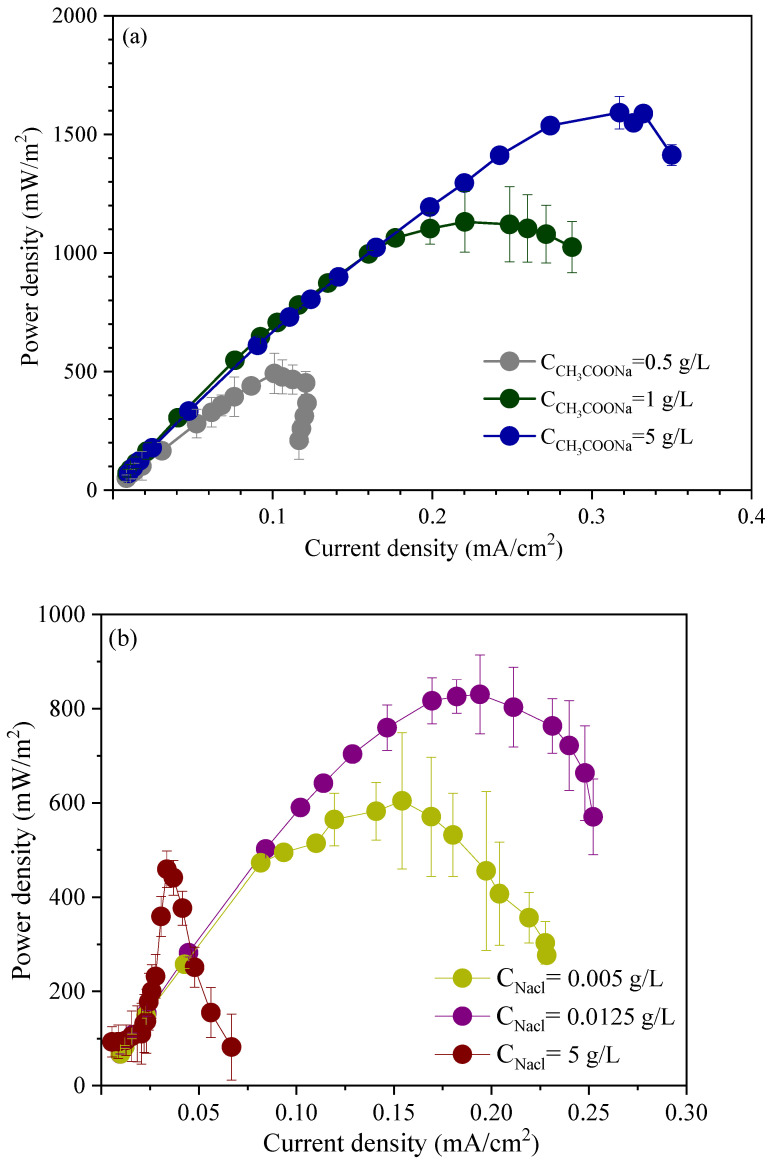
Electrochemical performance of MFCs under different water quality conditions. (**a**) power density at acetate concentration of 0.5 g/L, 1 g/L, and 5 g/L; (**b**) power density at sodium chloride concentration of 0.005 g/L, 0.0125 g/L, and 5 g/L; (**c**) CV curves of 0.5 g/L, 1 g/L, and 5 g/L acetate concentration; (**d**) CV curves of 0.005 g/L, 0.0125 g/L, and 5 g/L sodium chloride concentration; (**e**) oxidation reduction potential curves of 0.5 g/L, 1 g/L, and 5 g/L acetate concentration; (**f**) oxidation reduction potential curves of 0.005 g/L, 0.0125 g/L, and 5 g/L sodium chloride concentration.

**Figure 3 microorganisms-13-02077-f003:**
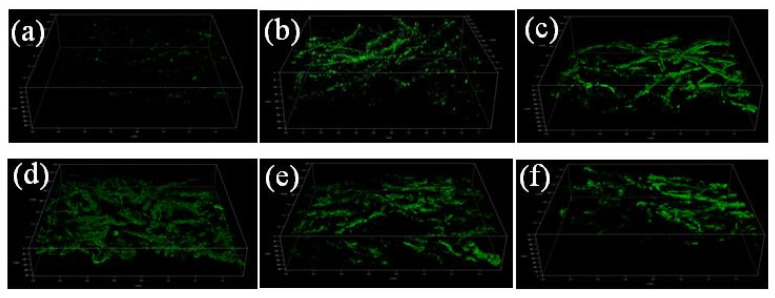
CLSM images under different water quality conditions. (**a**) Acetate concentration of 0.5 g/L; (**b**) acetate concentration of 1 g/L; (**c**) acetate concentration of 5 g/L; (**d**) sodium chloride concentration of 0.005 g/L; (**e**) sodium chloride concentration of 0.0125 g/L; (**f**) sodium chloride concentration of 5 g/L.

**Figure 4 microorganisms-13-02077-f004:**
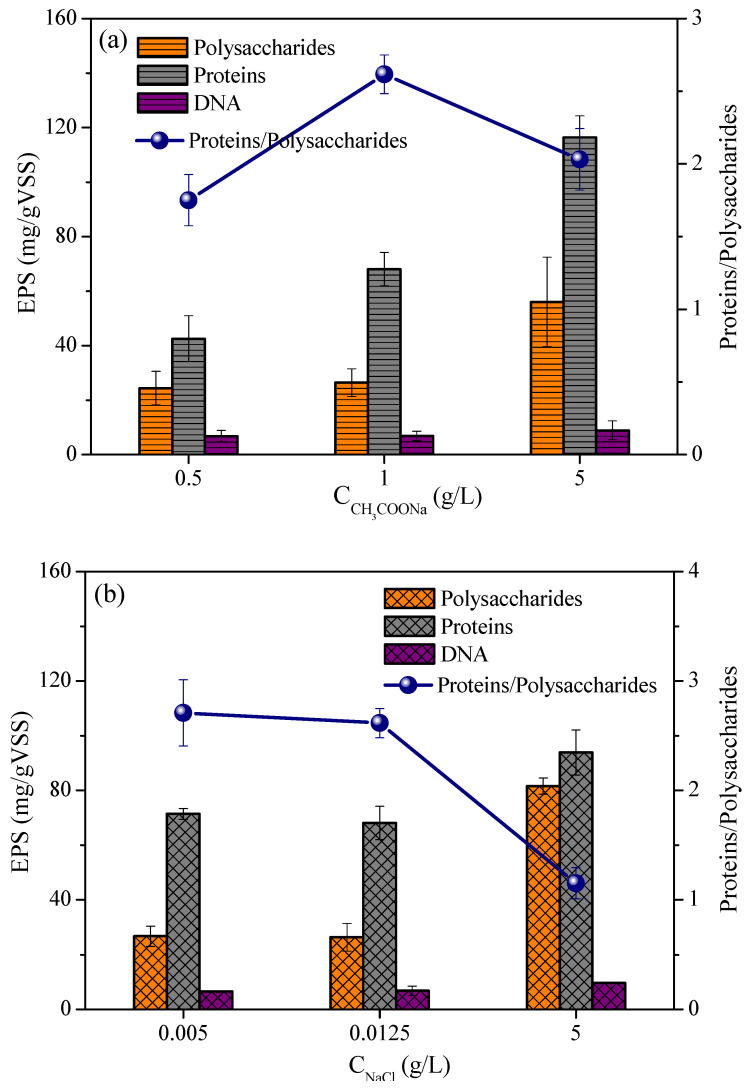
Influence on EPS under different water quality conditions. (**a**) Acetate concentration of 0.5 g/L, 1 g/L, and 5 g/L; (**b**) sodium chloride concentration of0.005 g/L,0.0125 g/L, and 5 g/L.

**Figure 5 microorganisms-13-02077-f005:**
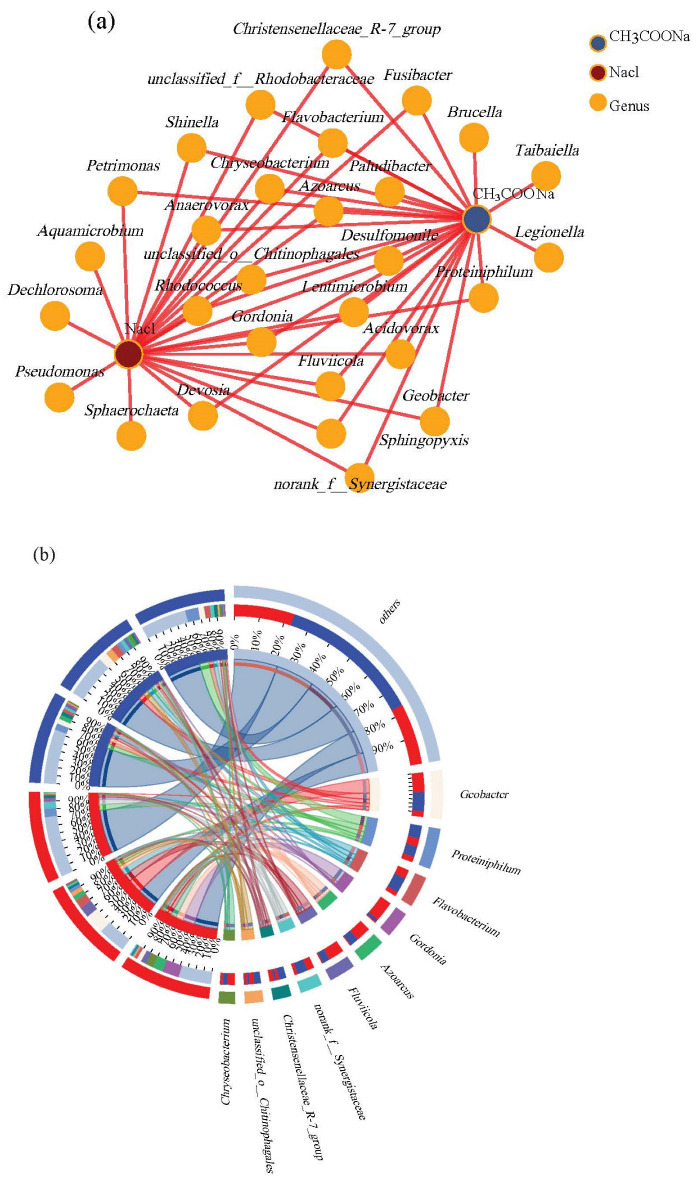
Microbial community with 2,4-DCP under different acetate concentration and sodium chloride concentration. (**a**) The co-occurrence network diagram of different acetate concentration and sodium chloride concentration; (**b**) circossample and species relationship diagram of different acetate concentration and sodium chloride concentration; (**c**) principal component analysis on genus level of different acetate concentration and sodium chloride concentration.

## Data Availability

The original contributions presented in this study are included in the article/Appendix A. Further inquiries can be directed to the corresponding author.

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
