# Peer review of "The Influence of Acetate and Sodium Chloride Concentration on the Toxic Response of Electroactive Microorganisms"

_microorganisms, 2025, doi:10.3390/microorganisms13092077_

Round 1
Reviewer 1 Report
Comments and Suggestions for Authors
The article presents the influence of feed concentration and salinity on the detection of toxic compounds in a microbial fuel cell. These are interesting issues for researchers focused on this application of microbial fuel cells.
1) However, when one begins to read the paper—starting with the title and abstract—it is difficult to understand the objective of the study. I recommend modifying the title and the beginning of the abstract to clarify that the study does not address the influence of the water quality, which is too broad, but rather the influence of feed concentration and salt concentration.
The sentence "For acetate, the evaluation results reliable with 1 g/L." needs to be better expressed and clarified in the abstract.
2) In this sentence "However, industrial wastewater often contains a large amount of toxic and harmful pollutants, which can inhibit microorganisms[1] in the treatment system[2], cause deterioration of effluent quality, and threaten water ecological security" at the introduction., I believe it would be useful to mention wastewater containing refractory chemical compounds that have been previously treated using microbial fuel cells. Some examples can be found in the following references:
-
Galai, S., Perez de los Rios, A., Hernández‐Fernández, F. J., Haj Kacem, S., Mateo Ramírez, F., & Quesada‐Medina, J. (2015). Microbial fuel cell application for azoic dye decolorization with simultaneous bioenergy production using Stenotrophomonas sp. Chemical Engineering & Technology, 38(9), 1511–1518. https://doi.org/10.1002/ceat.201500208
-
Addi, H., Mateo-Ramírez, F., Ortiz-Martínez, V. M., Salar-García, M. J., Hernández-Fernández, F. J., Perez de los Rios, A., Godinez, C., Lotfi, E. M., El Mahi, M., & Lozano Blanco, L. J. (2018). Treatment of mineral oil refinery wastewater in microbial fuel cells using ionic liquid based separators. Applied Sciences, 8(3), 438. https://doi.org/10.3390/app8030438
-
Luo, Y., Zhang, R., Liu, G., Li, J., Qin, B., Li, M., & Chen, S. (2011). Simultaneous degradation of refractory contaminants in both the anode and cathode chambers of the microbial fuel cell. Bioresource Technology, 102(4), 3827–3832. https://doi.org/10.1016/j.biortech.2010.11.121
-
Luo, Y., Zhang, R., Liu, G., Li, J., Li, M., & Zhang, C. (2010). Electricity generation from indole and microbial community analysis in the microbial fuel cell. Journal of Hazardous Materials, 176(1–3), 759–764. https://doi.org/10.1016/j.jhazmat.2009.11.100
"In most studies, the concentration of acetate as the substrate for MFCs was 1 g/L[17]. When the concentration of acetate was greater than 100 mM, the current of MFCs no longer increased, and the substrate reached saturation concentration, which conformed to the growth kinetics of microbial saturated substrates[18, 19]. A marine microbial fuel cell cultured with mixed bacteria, according to the Monod equation, had a current density close to a constant value when the concentration of acetate was greater than 0.50 mM [20]. There have been many reports on the effect of acetate concentration about electricity production, but there were few reports on its response to biological inhibition."
the different ways of expressing acetate concentration should be unified. Moreover, the final part should be made clearer to indicate that the objective is to study the influence of acetate concentration on the inhibitory response to certain contaminants. 4) The objective of the study needs to be clarified more clearly in the following sentence:
"The existing research results mainly focus on the electricity production of electroactive microorganisms and cannot be directly applied to the biological inhibition evaluation of wastewater. In order to make the evaluation results of electroactive microbial inhibition relatively accurate, reliable,"
which, in my opinion, is to evaluate the influence of acetate and NaCl concentrations on the inhibitory response of a microbial fuel cell during the treatment of contaminated aqueous effluents. 5) Figure S1 shows images under different acetate and sodium chloride concentrations. It is not clear what these images refer to — are they showing the anodes? This should be specified both in the figure and in the text. 6) In the sentence "For0.0125 g/L sodium chloride" plesae include a space. Please revise spaces between words in others sentence along paper. 7)A very important aspect that needs to be clarified is why, starting from section 3.2, the effect of the toxic compound is no longer studied. Instead, the effects of acetate and salt concentrations are analyzed, ignoring the influence of the toxicant. The authors should clearly explain why they took this approach and how this second part is related to the title of the paper. Could ne interesting to know the influence of the toxicant on the paremeters analyses from section 3.2. I would like to thank the author for the clarification Comments on the Quality of English Language
The English is generally correct, but some sentences should be improved to enhance clarity and understanding.
Author Response
|
Comments 1:The article presents the influence of feed concentration and salinity on the detection of toxic compounds in a microbial fuel cell. These are interesting issues for researchers focused on this application of microbial fuel cells. However, when one begins to read the paper—starting with the title and abstract—it is difficult to understand the objective of the study. I recommend modifying the title and the beginning of the abstract to clarify that the study does not address the influence of the water quality, which is too broad, but rather the influence of feed concentration and salt concentration. The sentence "For acetate, the evaluation results reliable with 1 g/L." needs to be better expressed and clarified in the abstract. |
|
Response 1: We really appreciate your precious advice, which could be a big help for the paper’s improvement. The title was revised as ”the influence of acetate and sodium chloride concentration on toxic response of electroactive microorganisms” The abstract was revised as follows: This study discussed the influence of acetate and sodium chloride concentration on monitoring 2,4-dichlorophenol(2,4-DCP) by electroactive bacteria. The performance of the reactor was represented by power density, and the electrochemical activity was represented by redox capacity. At the same time, microelectrodes were used to detect the redox potential between biofilms, and the changes in extracellular polymers and microbial community structure under different conditions were also explored. With acetate concentration of 1 g/L and sodium chloride concentration of 0.0125 g/L, the electroactive microorganisms were more sensitive to toxic substances and respond fast. The biofilm also evenly covered on the surface of the carrier, which aided in the diffusion of substances. Although the maximum power density monotonically increased with acetate concentration. High concentration of substrate may mask the inhibitory effect and affect the judgment of inhibitory results. The content of protein and polysaccharide increased monotonically with sodium chloride concentration. However, more polysaccharides would lead to high resistance to electron transfer. Compared to sodium chloride, the microbial content was more affected by acetate. The electroactive microorganisms had strong adaptability to salinity. In practical application, it’s conducive for increase the sensitivity of MFCs to reasonably reduce the concentration of acetic acid and sodium chloride.
|
|
Comments 2: In this sentence "However, industrial wastewater often contains a large amount of toxic and harmful pollutants, which can inhibit microorganisms[1] in the treatment system[2], cause deterioration of effluent quality, and threaten water ecological security" at the introduction., I believe it would be useful to mention wastewater containing refractory chemical compounds that have been previously treated using microbial fuel cells. Some examples can be found in the following references: Galai, S., Perez de los Rios, A., Hernández‐Fernández, F. J., Haj Kacem, S., Mateo Ramírez, F., & Quesada‐Medina, J. (2015). Microbial fuel cell application for azoic dye decolorization with simultaneous bioenergy production using Stenotrophomonas sp. Chemical Engineering & Technology, 38(9), 1511–1518. https://doi.org/10.1002/ceat.201500208 Addi, H., Mateo-Ramírez, F., Ortiz-Martínez, V. M., Salar-García, M. J., Hernández-Fernández, F. J., Perez de los Rios, A., Godinez, C., Lotfi, E. M., El Mahi, M., & Lozano Blanco, L. J. (2018). Treatment of mineral oil refinery wastewater in microbial fuel cells using ionic liquid based separators. Applied Sciences, 8(3), 438. https://doi.org/10.3390/app8030438 Luo, Y., Zhang, R., Liu, G., Li, J., Qin, B., Li, M., & Chen, S. (2011). Simultaneous degradation of refractory contaminants in both the anode and cathode chambers of the microbial fuel cell. Bioresource Technology, 102(4), 3827–3832. https://doi.org/10.1016/j.biortech.2010.11.121 Luo, Y., Zhang, R., Liu, G., Li, J., Li, M., & Zhang, C. (2010). Electricity generation from indole and microbial community analysis in the microbial fuel cell. Journal of Hazardous Materials, 176(1–3), 759–764. https://doi.org/10.1016/j.jhazmat.2009.11.100
|
|
Response 2: We really appreciate your precious advice, which could be a big help for the paper’s improvement. P4 Line 132-134: It has been reported microbial fuel cells can treated wastewater containing refractory chemical compounds [3-6]. References were added as follows: 3. Galai, S., et al., Microbial fuel cell application for azoic dye decolorization with simultaneous bioenergy production using Stenotrophomonas sp. Chemical Engineering & Technology, 2015.38(9): p.1511–1518. 4. Addi, H., et al., Treatment of mineral oil refinery wastewater in microbial fuel cells using ionic liquid based separators. Applied Sciences, 2018. 8(3): p.438. 5. Luo, Y., et al., Simultaneous degradation of refractory contaminants in both the anode and cathode chambers of the microbial fuel cell. Bioresource Technology, 2011.102(4) : p. 3827–3832. 6. Luo, Y., et al., Electricity generation from indole and microbial community analysis in the microbial fuel cell. Journal of Hazardous Materials, 2010.176(1–3) : p.759–764.
|
|
Comments 3: In the following sentence: "In most studies, the concentration of acetate as the substrate for MFCs was 1 g/L[17]. When the concentration of acetate was greater than 100 mM, the current of MFCs no longer increased, and the substrate reached saturation concentration, which conformed to the growth kinetics of microbial saturated substrates[18, 19]. A marine microbial fuel cell cultured with mixed bacteria, according to the Monod equation, had a current density close to a constant value when the concentration of acetate was greater than 0.50 mM [20]. There have been many reports on the effect of acetate concentration about electricity production, but there were few reports on its response to biological inhibition." the different ways of expressing acetate concentration should be unified. Moreover, the final part should be made clearer to indicate that the objective is to study the influence of acetate concentration on the inhibitory response to certain contaminants. 4) The objective of the study needs to be clarified more clearly in the following sentence: "The existing research results mainly focus on the electricity production of electroactive microorganisms and cannot be directly applied to the biological inhibition evaluation of wastewater. In order to make the evaluation results of electroactive microbial inhibition relatively accurate, reliable," which, in my opinion, is to evaluate the influence of acetate and NaCl concentrations on the inhibitory response of a microbial fuel cell during the treatment of contaminated aqueous effluents. 5) Figure S1 shows images under different acetate and sodium chloride concentrations. It is not clear what these images refer to — are they showing the anodes? This should be specified both in the figure and in the text. 6) In the sentence "For0.0125 g/L sodium chloride" plesae include a space. Please revise spaces between words in others sentence along paper. 7)A very important aspect that needs to be clarified is why, starting from section 3.2, the effect of the toxic compound is no longer studied. Instead, the effects of acetate and salt concentrations are analyzed, ignoring the influence of the toxicant. The authors should clearly explain why they took this approach and how this second part is related to the title of the paper. Could ne interesting to know the influence of the toxicant on the paremeters analyses from section 3.2. I would like to thank the author for the clarification
|
|
Response 3: We really appreciate your precious advice, which could be a big help for the paper’s improvement. “In most studies, the concentration of acetate as the substrate for MFCs was 1 g/L[21].”was revised as “In most studies, the concentration of acetate as the substrate for MFCs was 12.2 mM [21].” “The existing research results mainly focus on the electricity production of electroactive microorganisms and cannot be directly applied to the biological inhibition evaluation of wastewater. In order to make the evaluation results of electroactive microbial inhibition relatively accurate, reliable, and stable, this study intends to study the biological inhibition response of electroactive microorganisms under different concentrations of acetate and sodium chloride in the biological anode.” was revised as “Therefore, it is necessary to conduct research on the the influence of acetate and sodium chloride concentration on the inhibitory response of a microbial fuel cell during the treatment of contaminated aqueous effluents.” “Figure S1 shows images under different acetate and sodium chloride concentrations.” was revised as “Images of anode under different acetate concentration of 0.5 g/L, 1 g/L, 5 g/L and sodium chloride concentration of 0.005 g/L, 0.0125 g/L, 5 g/L.” "For0.0125 g/L sodium chloride" was revised as “For 0.0125 g/L sodium chloride” . We have revised spaces between words in others sentence along paper. Section 3.1 found that the response of electroactive microorganisms to 2,4-DCP varies under different acetate and sodium chloride concentration. Therefore, section 3.2 began to explore the mechanism behind this situation. An analysis was conducted on the electricity production performance, electrochemical activity, redox ability between biofilms, secretion of extracellular polymers, and microbial conditions under different acetate and sodium chloride concentration. Section 3.2 was an exploration of the mechanism behind the phenomena in Section 3.1. |

Reviewer 2 Report
Comments and Suggestions for Authors
This manuscript examines how water quality parameters, specifically acetate and sodium chloride, affect electroactive microbial communities in detecting toxins. It combines electrochemical analysis, biofilm characterization, and microbial community studies to explore environmental influences on toxicity responses, which is relevant for wastewater treatment and biosensor development. The structure is logical, and experiments are thorough. However, improvements are needed in clarity, methodology details (such as the toxicant used and inhibition measurement), and language. Below are my comments on the MS.
1.-Abstract: Please make the abstract more reader-friendly and concise. It should clearly state the study’s scope, including the toxicant used (2,4-dichlorophenol), the experimental conditions, and a brief summary of key findings. Remove any unclear or incomplete sentences to improve clarity.
2.-Materials and methods: This section needs more details for clarity and reproducibility. Please clarify how "toxic response” or inhibition was measured, such as when 2,4-DCP was added, observation duration, and what defines an “inhibition response." Defining terms like “inhibition response time” and “current inhibition rate" would help. Also, include the number of replicates and statistical methods used, especially if data are shown as means ± SD. These details will strengthen the study.
3.-Results and Discussion:
-Comment 1: The manuscript has lots of data, but the interpretation needs to be clearer and more connected. The authors should explain why the best toxic response was at intermediate acetate (1 g/L) and sodium chloride (~0.0125 g/L).
-Comment 2: Currently, the authors mention that high substrate masks toxicity and extreme salinity stresses microbes, but adding more explanation or mechanisms would help. For example, they could discuss how high acetate might delay toxicity by supporting microbial activity or how low acetate slows response due to nutrient limits.
-Comment 3: Similarly, clarify how salinity affects cells; too low or too high salt reduces inhibition, but the reasons could be clearer.
-Comment 4: The microbial community data (Figure 5) is a strength, but needs clearer integration. Clarify what “positively correlated” means—whether it indicates similar taxa or shifts. Also, briefly explain the analyses (co-occurrence, circos, PCA) and relate findings to sensor performance, highlighting key taxa.
4.-Conclusions: The study shows that lower acetate and salt improve MFC sensor sensitivity, but extreme conditions can harm the community. Explaining how these findings apply to real wastewater, different toxins, or future research would make the conclusions more practical and forward-looking.
Minor issues:
-Revise awkward or incorrect phrases for clarity.
-The authors should ensure consistent use of the past tense when describing experimental results.
- Please use technical language: Replace "monotonously” with "monotonically" when describing increasing trends. Instead of saying "the effect of sodium chloride concentration on EPS... was great," specify “significant” or quantify the effect, like the 2.8-fold increase in polysaccharide content.
- Correct formatting issues: italicize microorganism names, add spaces after units and commas, and ensure proper figure references.
-Some figure descriptions need clarification, especially how inhibition rates are calculated and label readability (e.g., Figs. 1 and 5).
-Finally, the manuscript needs careful editing to improve clarity and correct grammar. A native English speaker or professional editor could help fix issues like subject-verb agreement.
Author Response
|
Comments 1:This manuscript examines how water quality parameters, specifically acetate and sodium chloride, affect electroactive microbial communities in detecting toxins. It combines electrochemical analysis, biofilm characterization, and microbial community studies to explore environmental influences on toxicity responses, which is relevant for wastewater treatment and biosensor development. The structure is logical, and experiments are thorough. However, improvements are needed in clarity, methodology details (such as the toxicant used and inhibition measurement), and language. Below are my comments on the MS. 1.-Abstract: Please make the abstract more reader-friendly and concise. It should clearly state the study’s scope, including the toxicant used (2,4-dichlorophenol), the experimental conditions, and a brief summary of key findings. Remove any unclear or incomplete sentences to improve clarity.
|
|
Response 1: We really appreciate your precious advice, which could be a big help for the paper’s improvement. The abstract was revised as follows: This study discussed the influence of acetate and sodium chloride concentration on monitoring 2,4-dichlorophenol(2,4-DCP) by electroactive bacteria. The performance of the reactor was represented by power density, and the electrochemical activity was represented by redox capacity. At the same time, microelectrodes were used to detect the redox potential between biofilms, and the changes in extracellular polymers and microbial community structure under different conditions were also explored. With acetate concentration of 1 g/L and sodium chloride concentration of 0.0125 g/L, the electroactive microorganisms were more sensitive to toxic substances and respond fast. The biofilm also evenly covered on the surface of the carrier, which aided in the diffusion of substances. Although the maximum power density monotonically increased with acetate concentration. High concentration of substrate may mask the inhibitory effect and affect the judgment of inhibitory results. The content of protein and polysaccharide increased monotonically with sodium chloride concentration. However, more polysaccharides would lead to high resistance to electron transfer. Compared to sodium chloride, the microbial content was more affected by acetate. The electroactive microorganisms had strong adaptability to salinity. In practical application, it’s conducive for increase the sensitivity of MFCs to reasonably reduce the concentration of acetic acid and sodium chloride.
|
|
Comments 2: 2.-Materials and methods: This section needs more details for clarity and reproducibility. Please clarify how "toxic response” or inhibition was measured, such as when 2,4-DCP was added, observation duration, and what defines an “inhibition response." Defining terms like “inhibition response time” and “current inhibition rate" would help. Also, include the number of replicates and statistical methods used, especially if data are shown as means ± SD. These details will strengthen the study.
|
|
Response 2: Thank you for your valuable suggestions and comments. “2,4-dichlorophenol(2,4-DCP) was added in the anode, then record the time. When the voltage changed by 5%, it is considered that electroactive microorganisms have responded to toxic substances. When the voltage change amplitude was less than 5%, it is considered that the toxic reaction has terminated. Inhibition reaction time referred to the period from the start of adding toxic substances to a voltage change amplitude of less than 5%. Inhibition rate refers to the difference in average current change between blank and adding toxic substances within the same reaction time.” was added.
|
|
Comments 3: 3.-Results and Discussion: -Comment 1: The manuscript has lots of data, but the interpretation needs to be clearer and more connected. The authors should explain why the best toxic response was at intermediate acetate (1 g/L) and sodium chloride (~0.0125 g/L).
|
|
Response 3: Thank you for your valuable suggestions and comments. We explained as follows: When the concentration of sodium acetate was 1 g/L, the electroactive microorganisms were more sensitive to toxic substances and respond fast. The maximum power density was high with 1131.3 ± 127.6mW/m2, and the peak current density was 5.6 mA/cm2. At 1.0 g/L, ORP decreased from -168.9 ± 3.5 mV to -209.3 ± 3.4 mV, and changed by 40 mV in the range of 200 μm. The distribution difference of ORP values between biofilms was relatively small, which can ensure the efficient utilization of organic matter. The biofilm also evenly covered on the surface of the carrier, which aided in the diffusion of substances. The ratio of protein to polysaccharide content was highest. Protein aided in transportation, while a small amount of polysaccharides exhibited low resistance to electron transfer. When the concentration of sodium chloride was 0.0125 g/L, the osmotic pressure of microorganisms was conducive to inhibiting response, the electroactive microorganisms were more sensitive to toxic substances and respond faster. The maximum power density was highest with 825.8 ± 35.6 mW/m2 and the peak current densities was 3.0 mA/cm2. At 0.0125 g/L, the distribution of ORP was also beneficial for electroactive microorganisms to exert their effects. The biofilm was evenly covered with a thin layer on the surface of carbon fiber, which beneficial for the diffusion of substances and accelerating the response to toxicity. The ratio of protein to polysaccharide content was highest, which led to low resistance to electron transfer and suitable osmotic pressure for microorganisms. |
|
Comments 4: 3.-Results and Discussion: Currently, the authors mention that high substrate masks toxicity and extreme salinity stresses microbes, but adding more explanation or mechanisms would help. For example, they could discuss how high acetate might delay toxicity by supporting microbial activity or how low acetate slows response due to nutrient limits.
|
|
Response 4: We really appreciate your precious advice, which could be a big help for the paper’s improvement. Acetate was converted into acetyl-CoA by acetyl-CoA synthase, which served as a key metabolic intermediate in the TCA cycle, promoting ATP production and carbon skeleton supply. With abundant nutrients, the ions of nutrients surrounding microorganisms were more than those of toxic substances, so microorganisms uptake more nutrients, thus masking the toxicity response. When acetate was insufficient, the production of acetyl-CoA decreased. The TCA cycle was disrupted, and the supply of oxaloacetic acid was insufficient, inhibiting energy production and amino acid synthesis. Microorganisms mainly devoted their energy to maintaining their normal growth and metabolism, thus their response to toxic substances slowed down.
|
|
Comments5: 3.-Results and Discussion: Similarly, clarify how salinity affects cells; too low or too high salt reduces inhibition, but the reasons could be clearer.
|
|
Response 5: Thank you for your valuable suggestions and comments. We explained as follows: The impact of salinity on cells was mainly reflected in osmotic pressure regulation, ion balance, and metabolic activity. Low or high salt concentrations may disrupt normal cell function, and toxicity testing at this time cannot fully reflect the inhibitory effect of toxic substances. In a low salt environment, when cells were in a hypotonic state, water continued to flow in, causing the cells to swell or even rupture. In high salt environments, cell dehydration can cause cytoplasmic wall separation, inhibit growth, or lead to death. And high concentrations of Na ⁺ and Cl ⁻ ions bind to cellular proteins, which can interfere with metabolic pathways and cause misjudgment of inhibitory responses.
|
|
Comments 6: 3.-Results and Discussion: The microbial community data (Figure 5) is a strength, but needs clearer integration. Clarify what “positively correlated” means—whether it indicates similar taxa or shifts. Also, briefly explain the analyses (co-occurrence, circos, PCA) and relate findings to sensor performance, highlighting key taxa.
|
|
Response 6: Thank you for your valuable suggestions and comments. It was revised as “Compared to sodium chloride, the microbial content was more affected by acetate. The electroactive microorganisms had strong adaptability to salinity.” The content of Geobacter, Flavobacterium, Fluviicola, Unclassified_o_chitinophagales increased first and then decreased with the concentration of sodium acetate. This was consistent with trend of the redox ability of MFC. The content of Proteinophilum, Norank_f_Synergistackeae, Christensenellaceae_R-7_group increased with the increase of sodium acetate concentration. This was consistent with the overall trend of power density changes exhibited of MFC.
|
|
Comments 7: 3.Conclusions: The study shows that lower acetate and salt improve MFC sensor sensitivity, but extreme conditions can harm the community. Explaining how these findings apply to real wastewater, different toxins, or future research would make the conclusions more practical and forward-looking.
|
|
Response7: Thank you for your valuable suggestions and comments. In the future, MFC sensors can be applied to the front end of biological treatment units in wastewater treatment plants to provide early warnings of wastewater toxicity. This can prevent high-concentration toxic wastewater from impacting the microorganisms in the biological treatment units, causing them to become paralyzed. Appropriate concentrations of acetate and sodium chloride can maintain better microbial activity and improve the accuracy of predicting wastewater toxicity. Therefore, this research has significant practical application value.
|
|
Comments 8: Minor issues: -Revise awkward or incorrect phrases for clarity. -The authors should ensure consistent use of the past tense when describing experimental results. -Please use technical language: Replace "monotonously” with "monotonically" when describing increasing trends. Instead of saying "the effect of sodium chloride concentration on EPS... was great," specify “significant” or quantify the effect, like the 2.8-fold increase in polysaccharide content. -Correct formatting issues: italicize microorganism names, add spaces after units and commas, and ensure proper figure references. -Some figure descriptions need clarification, especially how inhibition rates are calculated and label readability (e.g., Figs. 1 and 5).
|
|
Response 8: Thank you for your valuable suggestions and comments. -The awkward and incorrect phrases were revised in article in red. - The past tense was revised in article in red. -“The effect of sodium chloride concentration on polysaccharide was great.” was revised as “The content of protein and polysaccharide increased monotonically with the increase of sodium chloride concentration. For example, the content of polysaccharide in 5 g/L was 2.8 times higher than that in 0.005 g/L. The content of protein in 5 g/L was 1.3 times higher than that in 0.005 g/L.” -The formatting issues: italicize microorganism names, add spaces after units and commas, and ensure proper figure references were revised in article in red. - The figure descriptions were clarification as follows: “Figure 1. Profile of voltage with 2,4-DCP under different acetate concentration and sodium chloride concentration (a) acetate concentration of 0.5 g/L, 1 g/L, and 5 g/L; (b) sodium chloride concentration of 0.005 g/L, 0.0125 g/L, and 5 g/L; (c) the inhibition rates of 0.5 g/L, 1 g/L, and 5 g/L acetate concentration; (d) the inhibition rates of 0.005 g/L, 0.0125 g/L, and 5 g/L sodium chloride concentration.” was revised as “Figure 1. Profile of voltage with 2,4-DCP under different acetate concentration and sodium chloride concentration (a) acetate concentration of 0.5 g/L, 1 g/L, and 5 g/L; (b) sodium chloride concentration of 0.005 g/L, 0.0125 g/L, and 5 g/L; (c) the inhibition rates of 2,4-DCP under different acetate concentration; (d) the inhibition rates of 2,4-DCP under different sodium chloride concentration.” “Figure 5. Microbial community with 2,4-DCP under different water quality conditions. (a) the co-occurrence network diagram; (b) circossample and species relationship diagram; (c) principal component analysis on genus level.” was revised as ” Figure 5. Microbial community with 2,4-DCP under different acetate concentration and sodium chloride concentration. (a) the co-occurrence network diagram of different acetate concentration and sodium chloride concentration; (b) circossample and species relationship diagram of different acetate concentration and sodium chloride concentration; (c) principal component analysis on genus level of different acetate concentration and sodium chloride concentration.” |

Round 2
Reviewer 2 Report
Comments and Suggestions for Authors
The revised version of the MS warrants publication in its current form.